# Perspectives on healthy eating practices and acceptance of WIC-approved foods among parents of young children enrolled in WIC

**Nour M. Hammad**[1]*, **Melissa C. Kay**[2]

1 Department of Nutrition, Harvard University, Boston, Massachusetts, United States of America,
2 Department of Pediatrics, Duke University, Durham, North Carolina, United States of America

* nourhammad@fas.harvard.edu

## Abstract

### Introduction

The prevalence of childhood obesity remains high in the United States, particularly among children living in low-income households. Diet quality plays an important role in obesity prevention, particularly among mothers as they serve as role models. Those served by the Special Supplemental Nutrition Program for Women, Infants, and Children (WIC) receive nutrient-rich foods aimed at increasing diet quality, yet redemption is low. Digital interventions targeting WIC parents show potential for behavior change and could be used for childhood obesity prevention.

### Methods

This study describes the formative research conducted to understand perspectives on healthy eating practices, acceptance of WIC-approved foods, and preferences for the use of digital tools to improve the purchasing and consumption of WIC-approved foods to improve diet quality. In-depth interviews were conducted with 13 WIC parents and caregivers.

### Results

A variety of definitions for and misconceptions about healthy eating exist among WIC caregivers. Most purchased foods were fruits, vegetables, milk, cheese, and eggs and the least purchased foods were yogurt and peanut butter. The biggest facilitator for purchasing WIC-approved foods was the preference of children and caregivers, whereas the biggest barrier was children's picky eating behaviors. Most caregivers reported using their phone to get nutrition information. Most caregivers reported their interest in receiving weekly text messages and indicated preferences about receiving recipes.

### Conclusion

A text messaging program that includes sending weekly messages, recipes, and nutrition tips is hypothesized to improve diet quality and increase redemption of WIC-approved foods.

**Data Availability Statement:** Ethical restrictions around sharing our data include a small qualitative sample that could be re-identified and lack of prior approval from participants and the IRB to share

this data. We would be happy to share our summary matrices upon request, rather than the raw data. Here is a contact: Karen Carter, Senior IRB Board Specialist, Duke University Health System, Institutional Review Board, karen. carter@duke.edu.

**Funding:** This study was supported by the Duke Center for Research to Advance Healthcare Equity (REACH Equity), which is supported by the National Institute on Minority Health and Health Disparities under award number U54MD012530. The funders had no role in study design, data collection and analysis, decision to publish, or preparation of the manuscript. There was no additional external funding received for this study.

**Competing interests:** The authors have declared that no competing interests exist.

# Introduction

Obesity is a complex, multifaceted problem and is considered one of the most challenging public health issues worldwide. Nearly 20% of children in the United States (U.S.) have obesity, affecting 14.4 million children [1]. Childhood obesity puts children at a higher risk of developing metabolic syndrome, type 2 diabetes, kidney diseases, and cardiovascular diseases later in adulthood [2, 3]. Childhood obesity is also associated with stigma, depression, anxiety, lower self-esteem, school absenteeism and poorer academic performance [4–6]. These long- and short-term implications underscore the importance of preventing childhood obesity. Studies show that family-based interventions, especially those targeting mothers, can be facilitators of decreasing childhood obesity from the early stages of child development [7–9]. Mothers heavily influence their child's intake, as they are often gatekeepers of food in the home. They play a critical role in their children's eating behavior through role modeling, creating food associations, and providing access to certain types of food [10, 11].

The Special Supplemental Nutrition Program for Women, Infants, and Children (WIC) serves pregnant, postpartum, and breastfeeding women, infants, and children up to age 5 years who live in low-income households. WIC provides vouchers for healthy foods; nutrition education; breastfeeding support; and medical and social-service referrals [12]. A hallmark of the WIC program is the food packages which are designed to meet the specific nutritional needs of WIC beneficiaries and align with national dietary guidance. The food packages include vouchers for nutrient rich foods like fruits and vegetables, low-fat/skim milk, and whole-grain products. Studies show that participation in WIC positively influences diet quality and rates of overweight and obesity [13, 14]. However, WIC has experienced a decline in enrollment, retention, and food package redemption over the past decade [15, 16]. In addition to structural barriers associated with enrollment, dissatisfaction with the retail experience and confusion over which products are included contribute to decreased redemption of WIC-approved foods [17, 18]. However, little is known about how WIC participants view healthy eating and how WIC-approved foods are, or are not, embedded into the family diet. Given the impacts WIC has on childhood obesity and diet quality, efforts are needed to support the reach and satisfaction of WIC participants to maximize program efficacy.

Smartphone ownership is ubiquitous in the U.S. and particularly high among those living in low-income households [19–21]. Digital interventions that focus on parents' behaviors, specifically mothers, have proven to be efficacious and have a positive impact on childhood obesity [22–24]. Yet few have aimed to improve diet quality, particularly among low-income populations [25]. Thus, this study aims to describe findings from the formative research conducted to develop a digital behavioral intervention for parents and caregivers enrolled in WIC to improve maternal diet quality by supporting redemption of WIC-approved foods, and provide guidance for future intervention development [26].

# Materials and methods

The 21-item Standards for Reporting Qualitative Research (SRQR) checklist (S1 Checklist) was used to ensure transparency in conducting this research and reporting its results.

## a. Recruitment

Recruitment focused on WIC participants receiving benefits from a network of federally qualified health centers that serves medically vulnerable patients throughout central North Carolina. Specific details on the recruitment procedure are reported elsewhere [26]. Briefly, WIC parents and caregivers were recruited by WIC clinicians during their regularly scheduled appointments. WIC clinicians recorded contact information for interested WIC parents (i.e.,

participants) using an online study form securely stored in a password-protected Box folder that could only be accessed by the research team. The research team followed up with those who were interested and assessed eligibility. If eligible, the parent completed an online consent form and baseline survey and scheduled a date and time for the interview.

## b. Participants

Recruitment occurred from May 2020 through August 2020. Eligibility criteria included the following: parent or caregiver with a child 2 years old or younger, receiving benefits from WIC, have a cell phone that can receive text messages, and are English speakers. We enrolled 13 parents and caregivers of children receiving WIC benefits. A larger sample size was planned, but we faced significant recruitment challenges due to the COVID-19 pandemic. Though the sample is small, prior work demonstrates it is sufficient for understanding common perceptions and experiences among a relatively homogenous group [27]. The flow of participants through the formative research is presented in Fig 1.

## c. Procedures

We conducted in-depth, semi-structured interviews [28] to obtain detailed information on the redemption of WIC-approved foods and explore food preferences and habits [29, 30]. An interview guide was developed by MCK, the principal investigator, with input from stakeholders including WIC clinicians and content experts in qualitative research. The interview guide consisted of open-ended questions related to redemption and use of foods received in WIC packages (i.e., liked and disliked foods as well as least and most purchased food items); thoughts about and practice of healthy eating (i.e., participants' definition of healthy eating, their cooking methods, and their source of nutrition information); and WIC's role in promoting healthy eating. Other questions included thoughts about digital interventions, such as the helpfulness of receiving text messages in addition to preferences about the frequency of receiving text messages.

Digitally audio-recorded interviews (n = 13) were conducted between June and September 2020. The interviews were on average 25 minutes long (range 13 to 39 minutes). All interviews were conducted by trained researchers. Participants were compensated with a $25 Walmart gift card. All interviews were transcribed by a professional transcription agency and verified for accuracy. All study procedures were approved by the ethical review board at Duke University.

## d. Analysis

We used an in-depth reflexive thematic approach to data analysis to identify, analyze, and report themes within data [31]. Mostly structural codes were used to create a codebook containing 14 sub-themes clustered into four themes. NH and MCK each independently coded the interviews using NVivo12 [32]. Although inter-coder reliability is discouraged with the reflexive thematic approach, we conducted it to assess the need for improvement in the quality and credibility of the coding process [33, 34]. Any individual code that had a kappa below 0.81 [35] was reviewed as a team and every discrepancy was discussed and corrected. This was feasible given the small sample size. Agreement was met on every code and the final overall kappa was 0.96.

## Results

### a. Description of the sample

Most of the participants were the child's mother (n = 12); 1 was the grandmother (Table 1). Participants had a mean age of 31 years (range 23–45), approximately two thirds (n = 8)

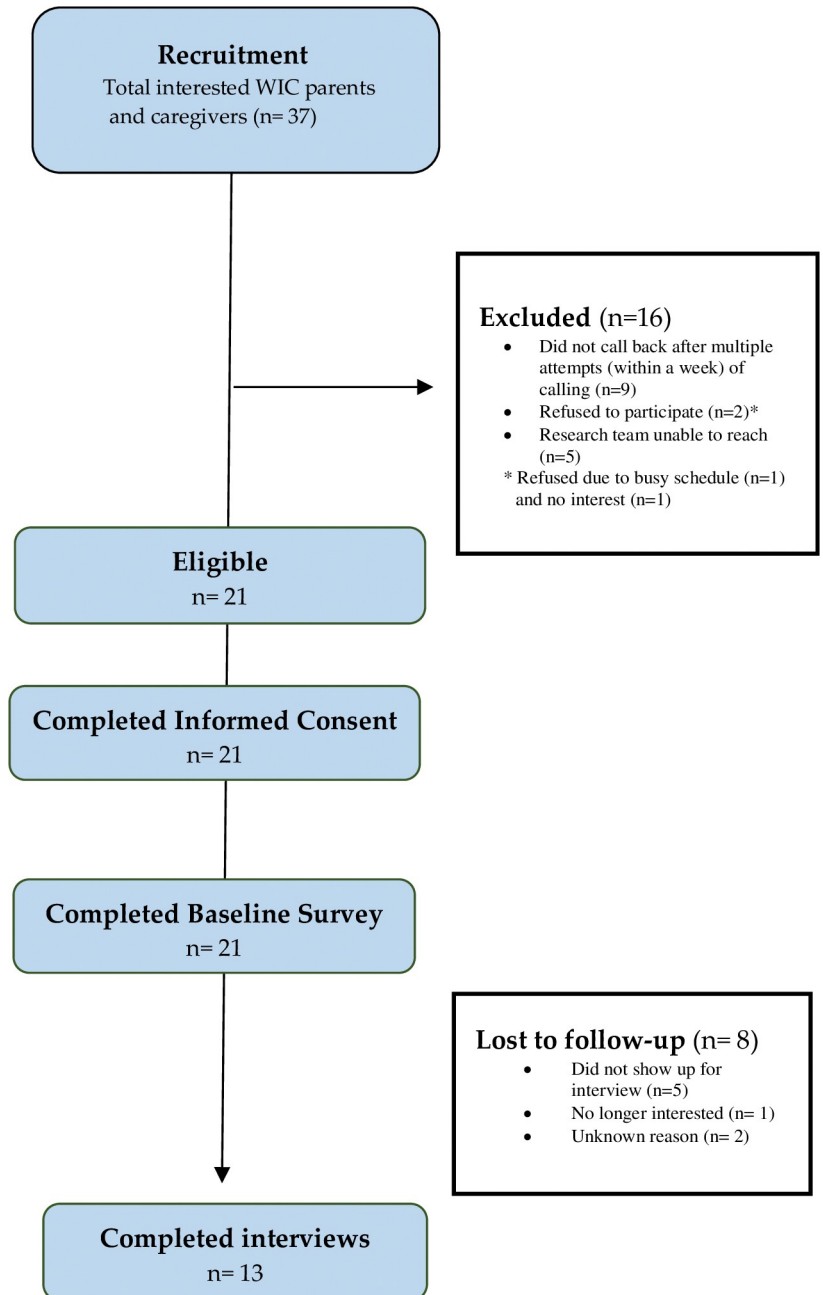

**Fig 1. Consort diagram describing flow of participants through the formative research.**

identified as White, with one identifying as White and Hispanic/Latina; the remaining five participants identify as non-Hispanic Black. About half (n = 7) had some college education or obtained a college degree, with five participants graduating from high school.

## b. Qualitative findings

The qualitative findings are presented below. Theme 1 focuses on healthy eating; theme 2 focuses on the purchase of WIC-approved foods; theme 3 focuses on WIC's role in promoting

**Table 1. Sociodemographic characteristics of the participants who completed interviews (n = 13).**

| Age, mean in years (range) | | 31.4 (23.0–45.0) |
|---|---|---|
| Race/Ethnicity, % (n) | Black | 38.5 (5) |
| | White | 53.8 (7) |
| | White and Hispanic/Latina | 7.7 (1) |
| Education, % (n) | Some high school | 7.7 (1) |
| | High school | 38.5 (5) |
| | Some college | 30.8 (4) |
| | ≥ College degree | 23.0 (3) |
| Married, % (n) | | 30.8 (4) |
| Employed*, % (n) | | 38.5 (5) |

*Both part-time (n = 2) and full-time employment (n = 3)

healthy eating; and theme 4 discusses participants' thoughts about a text messaging program designed to improve healthy eating, its content, and frequency of receiving text messages. Salient quotes were used to illustrate each sub-theme.

**Theme 1: (Un)healthy eating.** Participants defined their perspectives of healthy and unhealthy eating, their sources of nutrition information, and how they practice healthy eating.

*Healthy eating definitions and practices.* Healthy eating has different definitions according to different individuals. About half (n = 6) reported that daily consumption of fruits and vegetables encompasses healthy eating. Other definitions included portion control; moderation; balance and variety of foods when eating; eating fresh, unprocessed, organic and/or whole foods; eating proteins; whole grain foods; eating foods that help with digestion; and staying hydrated.

> *If you know how to control it and you don't intake too much junk food. It is not hard to have a healthy diet and enjoy some junk foods every once in a while.* (29-year-old, White mother)

About half (n = 6) reported trying to include fruits and vegetables with all their–and their children's–meals on a daily basis. Other non-frequently reported practices of healthy eating included finding healthier ways to prepare meals such as grilling and baking instead of frying, using cast iron pans, eating in moderation, following certain eating patterns such as intermittent fasting, eating organic foods, comparing food labels, eating fresh foods instead of canned or frozen foods, and sticking to routines.

> *Go to the event and if there's food there, you want to eat, eat it, but eat small portions of it.* (45-year-old, White grandmother)

*Unhealthy eating definitions and practices.* Participants were asked to specify definitions of foods or behaviors that are considered unhealthy. Most participants (n = 10) mentioned high sugar foods or foods containing added sugars as unhealthy, such as sodas, juice, and candies. As a 34-year-old Black mother said, "Things that I like... Like cake, ice cream, chocolate." According to participants, this definition of unhealthy eating contributes to negative health effects on the body like the effect of sugary beverages on one's dental health.

Many participants (n = 7) considered high sodium foods and processed foods to be unhealthy, such as canned foods, chips, macaroni and cheese, and hotdogs. Other examples of unhealthy food definitions included fast foods (n = 4), and overconsumption of any foods

(n = 2). One 45-year-old, White grandmother reported whole wheat foods as unhealthy, believing it raises blood glucose and insulin levels. Another participant reported foods that are labeled as "light" as unhealthy because of their aspartame content.

About half (n = 6) reported avoiding high salt and processed foods, frozen meals, and high sugar foods and beverages. Other practices included avoiding junk food, reading food labels, and diluting juices.

> *I thought that I was getting something healthy because it was a little snack bar for my son and then I looked at the sugar and it was like 25% of your daily value and it was just such a small little snack bar. And I was like, "Oh my gosh, this is really bad."* (27-year-old, White mother)

*Source of nutrition information.* Participants highlighted many different sources for getting information about nutrition, healthy eating, and meal preparation. Many participants (n = 7) reported obtaining their nutrition information from online resources such as Google, You-Tube, Pinterest, and Facebook. Some (n = 5) also referred to WIC nutritionists who would talk about healthy eating during their appointments or refer them to resources that help with meal preparation. Other cited recipes, friends and family, food labels, partners, cooking shows, and research. One participant also mentioned that she is her own source of information because she relies on her common sense.

*Devices used to get information about healthy eating.* When asked if they use their phone, computer, or tablet to get information about healthy eating, many (n = 7) mentioned using their phones for convenience and ease. Some (n = 4) mentioned using their computers, along with their phones, except for one participant who reported using her computer exclusively. Only one participant mentioned using a tablet, along with her phone and computer.

> *I just use my phone for everything. . . Yeah, it's just easier, it's right in your hand.* (29-year-old, White mother)

**Theme 2: Purchase of WIC-approved foods.** Participants reported the least and most purchased WIC-approved foods, and the barriers and facilitators, respectively, affecting these purchasing behaviors.

*Most purchased foods and facilitators to these purchases.* Fruits, vegetables, milk, cheese, and eggs were the most purchased WIC foods as presented in the order in Table 2.

The most cited reason for purchasing these foods was that the foods are used or liked by the children and/or themselves. Other reasons included that the foods are healthy, tasty, and free, and that there are many options that are provided such as lactose free milk and soy milk.

**Table 2. Most purchased WIC-approved foods (n = 13).**

| Foods most purchased | Number of participants who reported these foods |
| --- | --- |
| Fruits | 12 |
| Vegetables | 11 |
| Milk | 11 |
| Cheese | 11 |
| Eggs | 8 |
| Bread | 8 |
| Cereal | 7 |
| Peanut butter | 6 |
| Juice | 5 |
| Yogurt, Fish, Beans | <5 |

**Table 3. Least purchased WIC-approved foods (n = 12).**

| Foods least purchased | Number of participants who reported these foods |
|---|---|
| Yogurt | 6 |
| Peanut butter | 3 |
| Bread | 3 |
| Cereal, Beans, Fish, Milk, Juice, Eggs, Brown Rice, Canned Foods, Fruits, Vegetables | <3 |

*Least purchased foods and barriers to these purchases.* Yogurt and peanut butter were reported as the least purchased foods as presented in Table 3.

The biggest barrier for purchasing WIC-approved foods was that their children are picky eaters and dislike the foods (n = 8). Only 2 participants reported not purchasing WIC-approved foods because they dislike those foods. Other barriers included the low amount of money allocated for purchasing fruits and vegetables; small in-store labeling; big food sizes leading to food waste; children's health conditions; limited options of WIC-approved foods; the negative health effects of some of WIC-approved foods (e.g., juice); and the unfamiliarity to cook certain foods.

*Sometimes I have trouble figuring out what brands of tortillas and other whole wheat stuff places are participating in WIC with because only Food Lion will put up the WIC signs and I haven't seen WIC signs in other stores. Like, Walmart is closer to us and they don't have their signs.* (27-year-old, White mother)

**Theme 3: WIC's role in healthy eating.** When asked about the role WIC plays in promoting healthy eating, participants presented a variety of ideas as well as how WIC helps with other matters.

*WIC's helpfulness in promoting healthy eating.* All participants believed that WIC helps them eat healthier because WIC provides only healthy foods in their food packages.

*It gives access to some healthy items that we need daily.* (29-year-old, White mother)

Another reason reported was that WIC nutritionists are nice and helpful; they are a great source for credible nutrition information about healthy eating and they sometimes help refer participants to other resources that can provide them with recipes, snack ideas, and physical activity recommendations.

*I've never met anybody that was not that nice and did not sit there and listen to me and hear me out.* (29-year-old, White mother)

The third most cited reason was WIC's resources, including educational pamphlets, shopping guide, MyPlate, and food models, which help with portion control. Other reasons included that WIC allows participants to try new food items such as soy milk and lactose-free milk that they wouldn't buy otherwise.

*Yes [it's helpful], because it [WIC] gives you a lot of information and a lot of resources if you have questions.* (29-year-old, Black mother)

Two participants shared that although they consider WIC to be helpful in promoting healthy eating, they think WIC can do better. That is because although WIC offers healthy foods, they do not offer guides to help them cook or use these foods. As one 34-year-old, Black mother shared, nothing can help one eat healthy unless they are willing to: "It's just a mind over matter thing for me as far as how I'm going to eat healthy because I can eat healthy without WIC or with it..." Another participant believed that WIC does not offer healthy foods as she believes that whole wheat foods are not healthy.

*Other benefits of WIC.* Few participants (n = 3) mentioned that WIC is helpful in ways others than promoting healthy eating. Some noted that WIC helps by providing formula which is expensive, breastfeeding classes, and ways to address their children's picky eating or feeding transitions.

*That [Formula] is definitely a blessing to me because a can of milk is expensive... But also the lady [WIC nutritionist] that I go to, she gives me like different eating habits and when to know how to transition from the bottle to the sippy cup. So, those are helpful.* (34-year-old, Black mother)

*WIC's material vs general online material.* Six participants commented on the comparison between the material WIC offers (e.g., its website, physical material such as brochures, and appointments with WIC nutritionists) and online material (e.g., Google). Two of these participants could not make a comparison, because they do not use WIC's material.

*I haven't really explored it [WIC's website] to even understand it may have recipes on there or it may have other things that I could use for dinner.* (34-year-old, Black mother)

Two believed that WIC is more helpful because you can get answers during your appointment with WIC nutritionists and because WIC focuses more on healthy food items compared to online material.

*I guess some of the stuff that a nutritionist does, I could also just look up online on different blogs for toddlers and babies. So it's not really that different but it's a little more helpful to have a conversation with someone.* (27-year-old, White mother)

Two believe that online information is better, because although WIC provides healthy foods, online sources provide information on how to utilize them.

*They give me just the staple foods, whereas online would give me a little bit more information on how to make things, portion sizes, stuff like that.* (23-year-old, White mother)

*Ways WIC can help.* Five participants said they wish WIC could provide them with more money for fruits and vegetables to help them last for the entire month. Of note, during the pandemic, but after these interviews were collected, WIC increased the fruits and vegetables benefit from $7–11 to $24–47 per month [36].

*With my WIC package, I only get $9 for fruits and vegetables. And I mean, in this day and age, that's not really a lot to do with what fruits and vegetables you can get.* (27-year-old, White mother)

About half (n = 5) also said that WIC can help by providing more food options such as almond milk or vegetable smoothies and healthier cereals. Other wishes were that WIC could

send recipes to parents, update their pamphlets, offer organic options, focus more on parents, offer smaller sizes of foods, offer proteins such as chicken, and instruct stores to have bigger labels.

> *WIC could offer some type of weekly or monthly healthy recipe little sheet or something. Maybe once a month, they could pick up a recipe and send it out through emails or something.* (29-year-old, White mother)

> *I think it's a waste, because we don't eat a whole lot of yogurt, so it just sits in my refrigerator and then goes bad I throw it out . . . They have a one size fits all deal, and that just doesn't work when you're talking about food and different people's body types and what their health condition and what's going on in their bodies.* (45-year-old, White grandmother)

**Theme 4: Text messaging program to improve healthy eating.** When discussing a text messaging program aimed at improving healthy eating, participants generally had positive feedback.

*Thoughts about the program.* Most (n = 11) believed that developing a text messaging program aimed to improve intake is a good idea, as long as it provides non-repetitive information and gives participants the option to opt out.

> *Maybe they work from home and they use their phone for work. And if they're already getting a lot of emails and texts about work or something else, they may feel like it's overwhelming, but I think it would be a good idea.* (39-year-old, Black mother)

Two participants compared receiving text messages to emails and believe that receiving texts is easier than emails.

> *A lot of times people don't check their emails so I feel the texts would be more accessible.* (27-year-old, Black mother)

*Content of the program.* Most (n = 8) wanted recipes and some specifically mentioned healthy, inexpensive, and quick recipes or recipes for foods in season. The second most common content reported was tips and nutrition suggestions for both kids and parents (n = 6). Other ideas included giving examples of health benefits of foods provided in WIC packages, healthy eating habits that parents can follow, portion control, moderation, balance in meal preparation, motivational messages, research, and anything that is considered informational.

> *Because sometimes with a limited selection of things, it's hard to come up with new ideas of what to make with the food.* (27-year-old, White mother)

*Frequency of receiving text messages.* When asked about the frequency of the text messages, most (n = 10) preferred to receive text messages once a week, as frequencies more than once might get overwhelming. Around half (n = 5) suggested receiving text messages in the morning, which could serve as a reference if they go grocery shopping later or to get ideas for what to cook for dinner. They can also read the messages while they're on their way to work.

> *Just because I'm up. I have to work. So, I am constantly doing things. I may not have time to even talk, but I can look at my phone and read an article ahead of time versus at night, taking care of her. It's more of a downtime.* (34-year-old, Black mother)

*So I have time to think about during the day and then I'd be able to think about a dinner plan.* (23-year-old, White mother)

## Discussion

Interviewing a sample of caregivers of young children enrolled in WIC provided perspectives on healthy eating practices, preferences of WIC-approved foods, and ways WIC can improve support for healthy eating that can be used to guide future programs. The results demonstrate a variety of definitions for healthy (and unhealthy) eating that exist among caregivers participating in WIC. These definitions also contribute to individuals' practices of healthy eating as supported by previous research [37, 38]. However, surprisingly, in our sample only about half of participants reported eating fruits and vegetables as part of healthy eating practices. Results also highlight a variety of sources used to find nutrition information. These results confirm the need to understand perspectives on healthy eating to help in creating and delivering messages to inform behavior change [38, 39].

Uncovering the most and least purchased WIC-approved foods helps determine what WIC foods should be focused on to increase redemption rates and maximize program efficacy while developing the intervention. The most purchased food items identified were fruits, vegetables, milk, cheese, and eggs. The least purchased food items were yogurt and peanut butter. Our results are in alignment with a study conducted in 2018 that showed foods redeemed the most were milk, fruits, vegetables, and eggs and the foods redeemed the least included, beans, peanut butter, and whole grains [40]. Although our study had a smaller sample size, we identified cheese as another often purchased food and yogurt as one of the least purchased foods as opposed to whole grains. Using these results, future studies can focus on those foods participants are least likely to redeem by providing recipes and creative uses for such foods in meal and snacks throughout the day.

The biggest facilitator for purchasing WIC-approved foods was the preference of children and caregivers. However, the biggest barrier was children's picky eating behaviors. This underscores the importance of addressing this reoccurring child behavior. An expert panel report by the Healthy Eating Research national program suggests following these approaches to help address picky eating: parental modeling of healthy consumption, availability of healthy foods in the house, having family meals, and promoting food acceptance through repeated exposures to foods [41]. WIC can help by providing anticipatory guidance and education about these barriers and how to overcome them via educational material, communication during appointments with WIC nutritionists, and raising awareness about these barriers with providers who refer people to WIC.

Another barrier to full redemption of WIC-approved foods identified in this study is caregivers' dislike of some foods. Research has shown that the lack of flexibility in food packages to match individual preferences often results in benefits left unspent [42]. Other barriers include small in-store food labels, receiving excess amounts of foods, and unfamiliarity with certain foods' preparation. These findings align with those identified in recent studies [42, 43]. Another barrier to be considered is in-store grocery shopping during the COVID-19 pandemic; online grocery shopping has increased tremendously as a way to decrease transmission and adhere to safety precautions [44, 45]. However, there is no equitable access to online grocery shopping for WIC participants and structural changes to the WIC program are needed to make it more accessible to participants and optimize their shopping experience [46, 47].

WIC can further promote healthy eating by allocating more money towards purchasing fruits and vegetables and providing a larger variety of foods. The American Rescue Plan Act

enacted in March 2021 includes investments in WIC by temporarily increasing the participants' benefits for the purchase of fruits and vegetables in order to increase access to them which aims to improve the health, food security, and financial stability of WIC participants during the COVID-19 pandemic [48]. Future research should assess the perceived value of this increase in access to fruits and vegetables and expansion of WIC-approved food products. Furthermore, as a COVID-19 emergency response, WIC enacted some program flexibilities and provided its participants with COVID-19 food package waivers, allowing them to purchase a wider variety of foods [49]. Future studies should assess how such flexibilities impact the purchasing and healthy eating behaviors of participants [50].

Most caregivers reported using their phone to get nutrition information. Thus, a text messaging program could be used to improve healthy eating and WIC food redemption as it is well accepted by individuals. Similarly, in another study, texting was the most referenced feature used by WIC participants [51]. Text messaging interventions been successful in improving early post-delivery contact and exclusive breastfeeding rates among WIC mothers who receive peer counselling [52]. Our results suggest future interventions should consider using text messages that include recipes and nutrition tips to support WIC participants in making healthy behavior changes and improving the redemption of WIC-approved foods.

The strength of this study resides in the duality of coding which underscores the credibility of the results presented, leaving less room for error. This study also has limitations. First, two clinical research coordinators were responsible for conducting most of the interviews, which can add some inconsistency in data collection. The small sample size of 13 participants may offer less generalizable results; most participants came from urban settings and immigrant groups were not represented as the study was conducted in English only due to staffing and budget restrictions.

## Conclusions

The results of the formative research show the importance of communicating with the target audience before implementing an intervention to ensure that it is relevant and targeted to their needs. These findings can be coupled with previously published formative research to suggest an intervention aimed at improving WIC caregivers' healthy eating habits and improving redemption of WIC approved foods should include text messages with recipes and healthy eating tips. The findings also help inform ways to address barriers to redemption and improve caregivers' diet quality, which include tailoring educational material to these topics, raising awareness about these topics with WIC clinicians, and making structural changes to enable WIC participants to purchase WIC-approved foods online.

## Supporting information

**S1 Checklist. 21-item Standards for Reporting Qualitative Research (SRQR) checklist.** (DOCX)

## Acknowledgments

We thank Jennifer Headley and Dr. Mary Story at Duke University for their valuable input.

## Author Contributions

**Conceptualization:** Nour M. Hammad, Melissa C. Kay.

**Formal analysis:** Nour M. Hammad, Melissa C. Kay.

**Writing – original draft:** Nour M. Hammad.

**Writing – review & editing:** Melissa C. Kay.

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
