## [Decision Letter · Decision Letter 0]

29 Nov 2023

PONE-D-22-35532Perspectives on healthy eating practices and acceptance of WIC-approved foods among parents of young children enrolled in WICPLOS ONE

Dear Dr. Nour Hammad,

Thank you for submitting your manuscript to PLOS ONE. After careful consideration, we feel that it has merit but does not fully meet PLOS ONE’s publication criteria as it currently stands. Therefore, we invite you to submit a revised version of the manuscript that addresses the points raised during the review process.

We look forward to receiving your revised manuscript.

Kind regards,

Takele Gezahegn Demie, MPH

Academic Editor

PLOS ONE

Journal Requirements:

 “M.K. received the Duke Center for Research to Advance Healthcare Equity (REACH Equity) award, which is supported by the National Institute on Minority Health and Health Disparities under award number U54MD012530. The funders had no role in study design, data collection and analysis, decision to publish, or preparation of the manuscript.”

Additional Editor Comments (if provided):

Dear Authors,

Thank for submitting your work to PLOS ONE journal for possible publication. Your manuscript has been reviewed and reviewers have been recommended to further improve the manuscript and its writing before consideration for publication.

Reviewers' comments:

Reviewer's Responses to Questions

**Comments to the Author**

1. Is the manuscript technically sound, and do the data support the conclusions?

Reviewer #1: Yes

Reviewer #2: Yes

2. Has the statistical analysis been performed appropriately and rigorously? 

Reviewer #1: Yes

Reviewer #2: Yes

3. Have the authors made all data underlying the findings in their manuscript fully available?

Reviewer #1: Yes

Reviewer #2: Yes

4. Is the manuscript presented in an intelligible fashion and written in standard English?

Reviewer #1: Yes

Reviewer #2: Yes

5. Review Comments to the Author

Reviewer #1: The manuscript shows an important topic with regards to diet quality of parents and children who are beneficiaries of WIC program. This manuscript shows the perceptions of these caregivers toward the quality of food consumed. The manuscript describes in a qualitative way the data presented. The only limit is that the sample size was low.

Reviewer #2: Paper is very interesting expect same correction the paper is very interesting the result is well written however, in some parts it needs to get correction

Make classify your abstract into introduction ,methods, result, and conclusion part,

Don’t start sentences with problem at least you are expected to mention facts about your study like obesity and dietary practice

6. PLOS authors have the option to publish the peer review history of their article (what does this mean?). If published, this will include your full peer review and any attached files.

Reviewer #1: **Yes: **Maha Hoteit

Reviewer #2: **Yes: **Abera Lambebo

---

## [Author Response · Author response to Decision Letter 0]

30 Nov 2023

Reviewer #1: The manuscript shows an important topic with regards to diet quality of parents and children who are beneficiaries of WIC program. This manuscript shows the perceptions of these caregivers toward the quality of food consumed. The manuscript describes in a qualitative way the data presented. The only limit is that the sample size was low.

NH: We agree the sample size was smaller than desired. COVID significantly impacted our ability to recruit caregivers. The low sample size has been addressed as a limitation in the final paragraph in the discussion section.

Reviewer #2: Paper is very interesting expect same correction the paper is very interesting the result is well written however, in some parts it needs to get correction.

Abstract

Make classify your abstract into introduction ,methods, result, and conclusion part.

NH: The abstract has now been reformatted to fit into this classification. Additional information has been added under the results section. The abstract is now as follows: 

“Introduction: The prevalence of childhood obesity remains high in the United States, particularly among children living in low-income households. Diet quality plays an important role in obesity prevention, particularly among mothers as they serve as role models. Those served by the Special Supplemental Nutrition Program for Women, Infants, and Children (WIC) receive nutrient-rich foods aimed at increasing diet quality, yet redemption is low. Digital interventions targeting WIC parents show potential for behavior change and could be used for childhood obesity prevention. 

Methods: This study describes the formative research conducted to understand perspectives on healthy eating practices, acceptance of WIC-approved foods, and preferences for the use of digital tools to improve the purchasing and consumption of WIC-approved foods to improve diet quality. In-depth interviews were conducted with 13 WIC parents and caregivers. 

Results: A variety of definitions for and misconceptions about healthy eating exist among WIC caregivers. Most purchased foods were fruits, vegetables, milk, cheese, and eggs and the least purchased foods were yogurt and peanut butter. The biggest facilitator for purchasing WIC-approved foods was the preference of children and caregivers, whereas the biggest barrier was children’s picky eating behaviors. Most caregivers reported using their phone to get nutrition information. Most caregivers reported their interest in receiving weekly text messages and indicated preferences about receiving recipes. 

Conclusion: A text messaging program that includes sending weekly messages, recipes, and nutrition tips is hypothesized to improve diet quality and increase redemption of WIC-approved foods.”

Nearly 20% of children in the U.S. have obesity, affecting 14.4 million children [1]. 48 Childhood obesity puts children at a higher risk of developing metabolic syndrome, type 49 2 diabetes, kidney diseases, and cardiovascular diseases later in adulthood [2, 3]. 

Don’t start sentences with problem at least you are expected to mention facts about your study like obesity and dietary practice. 

NH: We added another statement to conceptualize the bigger picture of obesity. The introduction now reads: “Obesity is a complex, multifaceted problem and is considered one of the most challenging public health issues worldwide. Nearly 20% of children in the United States (U.S.) have obesity, affecting 14.4 million children [1]. Childhood obesity puts children…”

To ensure the intervention is congruent with the needs of WIC parents and 85 caregivers, we conducted in-depth interviews to guide the development. These interviews 86 focused on highlighting barriers and facilitators to consuming WIC-approved foods and 87 behaviors associated with healthy eating. The aim of this paper is to describe findings 88 from the formative research and provide guidance for future intervention development.

Make this part to method part it is not introduction part 

NH: We removed this section from the introduction and added a more appropriate sentence. 

The last sentence in the introduction now reads: “Thus, this study aims to describe findings from the formative research conducted to develop a digital behavioral intervention for parents and caregivers enrolled in WIC to improve maternal diet quality by supporting redemption of WIC-approved foods, and provide guidance for future intervention development [26].”

Table 2. Characteristics of the participants who completed interviews (n=13). Change to sociodemographic characteristics of the participants who completed interviews

NH: This table is now renamed: “Table 1. Sociodemographic characteristics of the participants who completed interviews (n=13).”

Paper is very interesting 

NH: Thank you

You should include a short title that is relevant to your topic

NH: We changed the short title to: “Eating practices and acceptance of WIC foods”.

please dont use abbreviations in "Abstract" (referring to US)

NH: Spelled out US in the first sentence in the abstract to United States.

no need please remove and insert the info inside the text (referring to Table 1)

NH: Deleted Table 1 and inserted its information into the text. Lines 129-136 now read: “The interview guide consisted of open-ended questions related to redemption and use of foods received in WIC packages (i.e., liked and disliked foods as well as least and most purchased food items); thoughts about and practice of healthy eating (i.e., participants’ definition of healthy eating, their cooking methods, and their source of nutrition information); and WIC’s role in promoting healthy eating. Other questions included thoughts about digital interventions, such as the helpfulness of receiving text messages in addition to preferences about the frequency of receiving text messages.”

Tables 2-4 are now tables 1-3.

---

## [Editor Report · Decision Letter 1]

4 Dec 2023

Perspectives on healthy eating practices and acceptance of WIC-approved foods among parents of young children enrolled in WIC

PONE-D-22-35532R1

Dear Ms. Nour Hammad,

We’re pleased to inform you that your manuscript has been judged scientifically suitable for publication and will be formally accepted for publication once it meets all outstanding technical requirements.

Within one week, you’ll receive an e-mail detailing the required amendments. When these have been addressed, you’ll receive a formal acceptance letter, and your manuscript will be scheduled for publication.

An invoice for payment will follow shortly after the formal acceptance. To ensure an efficient process, please log into Editorial Manager at http://www.editorialmanager.com/pone/, click the 'Update My Information' link at the top of the page, and double check that your user information is up to date. If you have any billing related questions, please contact our Author Billing department directly at authorbilling@plos.org.

Kind regards,

Takele Gezahegn Demie, MPH

Academic Editor

PLOS ONE

Additional Editor Comments (optional):

Thank you for revising the manuscript and your detailed response. Well improved! Congratulations!
---

## [Editor Report · Acceptance letter]

11 Dec 2023

PONE-D-22-35532R1 

Perspectives on healthy eating practices and acceptance of WIC-approved foods among parents of young children enrolled in WIC 

Dear Dr. Hammad:

I'm pleased to inform you that your manuscript has been deemed suitable for publication in PLOS ONE. Congratulations! Your manuscript is now with our production department. 

Kind regards, 

on behalf of

Mr. Takele Gezahegn Demie 

Academic Editor

PLOS ONE